# Research on a Metal Surface Defect Detection Algorithm Based on DSL-YOLO

**DOI:** 10.3390/s24196268

**Published:** 2024-09-27

**Authors:** Zhiwen Wang, Lei Zhao, Heng Li, Xiaojun Xue, Hui Liu

**Affiliations:** Faculty of Information Engineering and Automation, Kunming University of Science and Technology, Kunming 650032, China; 20222204239@stu.kust.edu.cn (Z.W.);

**Keywords:** surface defect detection, DWRB module, SADown module, LASPPF module

## Abstract

In industrial manufacturing, metal surface defect detection often suffers from low detection accuracy, high leakage rates, and false detection rates. To address these issues, this paper proposes a novel model named DSL-YOLO for metal surface defect detection. First, we introduce the C2f_DWRB structure by integrating the DWRB module with C2f, enhancing the model’s ability to detect small and occluded targets and effectively extract sparse spatial features. Second, we design the SADown module to improve feature extraction in challenging tasks involving blurred images or very small objects. Finally, to further enhance the model’s capacity to extract multi-scale features and capture critical image information (such as edges, textures, and shapes) without significantly increasing memory usage and computational cost, we propose the LASPPF structure. Experimental results demonstrate that the improved model achieves significant performance gains on both the GC10-DET and NEU-DET datasets, with a mAP@0.5 increase of 4.2% and 2.6%, respectively. The improvements in detection accuracy highlight the model’s ability to address common challenges while maintaining efficiency and feasibility in metal surface defect detection, providing a valuable solution for industrial applications.

## 1. Introduction

With the rapid advancement of industrial manufacturing technology, metal materials have become indispensable across numerous industries, serving as the backbone of critical components in various applications. However, the complexity of manufacturing processes introduces a myriad of potential defects, such as cracks, holes, and corrosion, on metal surfaces. These imperfections are not merely cosmetic flaws; they critically undermine the structural integrity and reliability of the products, leading to potential safety hazards, increased risk of equipment failure, and diminished production efficiency. Consequently, the precise detection and thorough analysis of metal surface defects are imperative for maintaining high standards in industrial production, ensuring product safety, and enhancing operational efficiency [1].

The advent of computer vision technology has revolutionized modern industry by enabling automated and more efficient quality control processes. In recent years, deep learning-based object detection algorithms have gained prominence due to their exceptional performance in image processing and computer vision tasks. These algorithms, powered by deep learning, can autonomously learn and extract intricate feature representations, making them particularly well-suited for identifying and classifying metal surface defects. The implementation of deep learning models in automated detection systems significantly reduces the need for manual inspection, thereby lowering labor costs. Moreover, these models can be seamlessly integrated into real-time production lines, enhancing production efficiency and ensuring consistent product quality [2]. The exploration of metal surface defect detection through deep learning not only plays a crucial role in elevating product standards and reducing manufacturing costs but also accelerates the digital transformation of the industrial sector, laying the foundation for smarter and more resilient manufacturing practices.

Target detection algorithms are generally divided into two categories: single-stage and two-stage detection algorithms. The single-stage target detection algorithm directly recognizes targets from images and has gained significant attention due to its excellent real-time adaptability. Among the classical single-stage detection algorithms, the YOLO [3,4,5] and SSD [6,7,8] series are particularly representative. Through innovative design, these algorithms transform the target detection task into a single regression problem, achieving an optimal balance between efficiency and detection accuracy. As models of single-stage object detection algorithms, the YOLO and SSD series excel not only in real-time performance but also in various complex scenarios. These algorithms provide highly accurate target location and recognition while maintaining fast response capabilities and have significantly contributed to the advancement of the target detection field. Moreover, single-stage target detection algorithms are widely used in various real-time applications because of their ability to simplify the computational process and reduce computational complexity.

In the field of metal surface defect detection, images often contain a large number of small-scale targets, with a single image potentially containing multiple defect features. The spatial distribution of these features is typically sparse. Additionally, due to insufficient illumination and dust interference common in industrial environments, metal surface images often exhibit blurriness. The redundant information caused by this blurriness can interfere with the feature extraction process of convolutional neural networks (CNNs). This interference not only complicates feature extraction but may also lead to the loss of fine-grained details and key features. These challenges weaken the model’s ability to learn features, ultimately limiting its performance in accurately identifying defect types and positions, which decreases detection accuracy. Current object detection models rely on complex network structures, but this complexity imposes a significant computational burden, limiting detection speed and causing high latency. To address these issues, we have designed an efficient, real-time detection model for metal surface defects called DSL-YOLO.

The purpose of this paper is to explore the research of metal surface defect detection algorithms based on the YOLO framework, aiming to achieve efficient and accurate automatic detection of metal surface defects using deep learning technology. The main contributions of this paper are as follows:Aiming at the problem of low accuracy and slow speed of metal surface defect detection, we propose a DSL-YOLO metal surface defect detection model based on YOLOv8. We have achieved a perfect balance between speed and accuracy.To address the issue that the core module C2f of YOLOv8 has limited extraction capabilities for small and occluded targets, weak ability to capture detailed information, and struggles to effectively extract sparse features distributed in space, we propose the C2f_DWRB structure.Ordinary convolutions in backbone networks may lead to the loss of fine-grained information and insufficient feature learning in challenging tasks involving blurred images or small objects. To enhance the model’s feature representation, we design the SADown structure.To further enhance YOLOv8’s multi-scale feature extraction capability and capture critical information, such as edges, textures, and shapes, while not significantly increasing memory usage and computational cost, we propose the LASPPF structure.

The remainder of this paper is organized as follows: In Section 2, we provide a detailed overview of the various methods currently used for metal surface defect detection and analyze their limitations. Section 3 will discuss the structure and working principles of the DSL-YOLO algorithm. In Section 4, we evaluate the feasibility and effectiveness of the algorithm in industrial applications through experimental results. Finally, Section 5 summarizes the paper and outlines potential directions for future research.

## 2. Related Work

Currently, most enterprises still depend on manual detection methods for industrial product defect identification [9]. However, the effectiveness of this approach is often significantly influenced by subjective factors, such as workers’ technical skills and working conditions, resulting in low efficiency.

With advancements in related technologies, various non-destructive testing methods have been widely developed for material testing, including magnetic particle testing, penetration testing, eddy current testing, ultrasonic testing, X-ray testing, and other defect detection techniques [10,11,12]. The rapid development of computer vision technology has led many researchers to apply image-based methods to material defect detection [13,14]. For instance, Song et al. [15] developed an anti-noise feature descriptor for identifying hot-rolled strip defects. This approach modifies the thresholding scheme by constructing a neighborhood evaluation window, thus enhancing both robustness and accuracy. Additionally, Seba et al. [16] proposed an unsupervised automatic defect texture detection model. This model utilizes non-generalized entropy with Gaussian gain as the regularity index, and texture blocks are locally computed using the sliding window method. The method introduces a dual-mode Gaussian mixture model to correct the entropy value and determines the detection window size based on the minimum entropy detection mode probability, achieving automatic defect detection without manual intervention. However, these methods often rely on manual feature extraction, which introduces subjectivity in feature selection and results in less satisfactory detection accuracy. Furthermore, many of these methods are limited to defect classification or are unable to perform more complex detection.

In recent years, significant progress has been made in deep learning-based object detection technologies, with numerous outstanding detection models being proposed and successfully applied to various detection tasks. These models include two-stage object detection networks represented by Faster R-CNN [17], as well as one-stage networks represented by YOLO and SSD. Additionally, Transformer models have garnered widespread attention [18]. For instance, Zhou et al. [19] improved the Fast R-CNN model to enhance its performance in detecting defects on steel strips. They combined a new residual spatial pyramid pooling module with a feature pyramid network to strengthen multi-scale feature fusion. Liu et al. [20] proposed a dual-branch network named CGTD-Net, which is based on a channel-global Transformer. At the end of the backbone network, Swin Transformer is employed, and a multi-channel feature pyramid network mitigates the negative effects of single-channel networks. Furthermore, the network incorporates a Spatial-Channel Global Attention (SCGA) module to further enhance the feature extraction capabilities for spatial and channel information. Zhou et al. [21] proposed a detection model named CABF-YOLO, which significantly improves the accuracy and efficiency of detecting defects on steel strips by introducing a triple convolutional coordinate attention module, a bidirectional fusion strategy, and an EIoU loss function.

In the field of metal defect detection, YOLO has demonstrated significant advantages due to its excellent detection speed and high accuracy. YOLO can process large amounts of image data in real-time, making it particularly suitable for application scenarios with high demands for real-time performance and accuracy, such as industrial production lines. Additionally, YOLO’s single-stage structure simplifies the detection process and reduces computational resource consumption, allowing it to perform well even in resource-constrained environments. In contrast, although two-stage object detection networks like Faster R-CNN exhibit superior detection accuracy, their complex network structure and slower detection speed limit their application in real-time detection tasks. While Transformer models enhance detection performance through more complex network structures and strong modeling capabilities, these models typically require substantial computational resources and memory, resulting in longer training and inference times, higher model complexity, and increased difficulty in deployment and maintenance.

Therefore, despite the potential of these emerging technologies to improve detection accuracy, their high computational costs and complexity still present certain limitations in practical applications. In contrast, YOLO’s fast and efficient characteristics make it particularly suitable for real-time tasks, such as defect detection, making it an important choice in industrial applications. Looking ahead, the integration of YOLO’s efficiency with the high precision of emerging technologies is expected to drive the object detection field toward greater efficiency and accuracy.

## 3. Methods

This section provides a detailed introduction to our baseline model and offers an in-depth discussion of the structure and functioning of the DSL-YOLO algorithm. We will also present the key proposed modules, including the DWRB (Dilation-wise Residual Block), SADown (Spatial Attention Downsampling), and LASPPF (Large Kernel Adaptive Spatial Pyramid Pooling Fast) modules. The design concepts, implementation details, and the role of each module in the model will be thoroughly analyzed to highlight their contributions to enhancing detection performance and efficiency. Through the integration of these modules, our model excels in handling complex defect detection tasks, particularly in terms of real-time performance and accuracy, achieving significant improvements in performance.

### 3.1. YOLOv8 Model

YOLOv8 is the latest model in the YOLO series, released by Ultralytics in January 2023. This series includes five versions, ranging from small to large: v8n, v8s, v8m, v8l, and v8x. As the model size increases, so does its detection accuracy. These different versions offer a range of options to address various tasks, including object detection, image classification, instance segmentation, and key point detection [22,23,24,25]. Users can select the appropriate model depth and width based on the complexity and accuracy requirements of specific tasks to achieve optimal performance.

In metal defect detection tasks, accuracy and real-time performance are the primary requirements. YOLOv8 offers a balance between recognition accuracy and detection speed, making it an efficient single-stage object detection algorithm. As the smallest model in the YOLOv8 series, YOLOv8n has notable advantages. Compared to other models in the YOLOv8 series, YOLOv8n significantly reduces model parameters and floating-point operations while maintaining adequate recognition accuracy. This makes YOLOv8n particularly suitable for object detection tasks with simple features and fewer categories, and it can deliver real-time detection performance even in resource-constrained environments.

The YOLOv8n model comprises four primary components: the input layer, backbone, neck, and head. The backbone is primarily composed of the C2f and CBS modules, which collectively extract deep features from the input image. The C2f module is designed to capture rich gradient information while maintaining the model’s lightweight nature, thereby enhancing feature representation capabilities. This design is particularly effective in capturing fine-grained features, giving the model an advantage in handling small objects or intricate textures. The CBS module integrates convolutional layers, batch normalization, and the Swish activation function, further improving the feature learning capacity and generalization performance of the model.

The neck structure incorporates both the Feature Pyramid Network (FPN) and the Path Aggregation Network (PAN) [26,27]. The FPN enhances the model’s ability to detect objects at various scales by generating feature maps at different resolutions. By combining high-resolution features from earlier layers with low-resolution features from deeper layers, the FPN provides a richer set of multi-scale features, thus improving detection performance for both large and small objects. The PAN aggregates features across different levels of the network, strengthening feature representation and enabling the model to better capture detailed information and global context.

The head consists of three detection branches of varying scales, with each branch dedicated to detecting objects at its respective scale. To optimize detection results, YOLOv8n employs the Non-Maximum Suppression (NMS) algorithm to eliminate redundant detections, ensuring the precision of the final output. This structural design allows YOLOv8n to excel in detecting targets in complex scenarios, effectively identifying and localizing objects of different sizes.

### 3.2. DSL-YOLOv8 Model Structure

The structure of the DSL-YOLO network is illustrated in Figure 1. Initially, the input image undergoes Mosaic data augmentation before entering the model’s network [28]. The backbone network includes our specially designed SADown and C2f_DWRB modules for feature extraction, which extract image features in different ways and generate feature maps. The extracted feature map is then fed into the LASPPF module, which enhances the model’s ability to focus on target areas through multi-scale feature concatenation and attention mechanisms [29,30]. The neck structure incorporates a Feature Pyramid Network (FPN) and a Path Aggregation Network (PAN) to fuse feature maps at different levels of the backbone network, enabling the capture of more comprehensive multi-scale information. Finally, the fused feature map is processed in the head module to produce the final detection result.

In the task of metal surface defect detection, we identified shortcomings in YOLOv8’s core module, C2f, particularly in handling small and occluded targets. Specifically, its ability to capture detailed information and extract sparsely distributed spatial features is limited. To address these issues, we propose the DWRB module as a replacement for the C2f Bottleneck structure in YOLOv8, designing the C2f_DWRB structure to more effectively extract detailed target information and sparse spatial features. When handling complex tasks involving blurred images or small objects, ordinary convolution often results in the loss of fine-grained information and insufficient feature learning. The extracted feature maps are input into the Neck network for further processing and fusion to optimize target detection performance. However, the original feature extraction capability is insufficient for effectively capturing multi-scale information, leading to inaccurate target predictions. To overcome this challenge, we designed the SADown structure to replace the standard convolution structure in the backbone network, thereby enhancing the model’s feature learning capability. Additionally, we replaced the convolution in the neck structure with SPD-Conv to improve the detection of small targets. To further enhance YOLOv8’s multi-scale feature extraction and its ability to capture key features (such as edges, textures, and shapes) of images while avoiding a significant increase in memory usage and computational costs, we propose the LASPPF structure. This structure optimizes feature extraction and enhances the model’s capacity to capture crucial details.

### 3.3. C2f _DWRB Structure

Currently, many models utilize Multi-rate Depthwise Separable Dilated Convolution to capture multi-scale context information from a single input feature map, enhancing feature extraction efficiency [31]. However, this approach often falls short in extracting sparsely distributed features and detailed information within images. To address this limitation, we designed a DWRB network, as illustrated in Figure 2, which optimizes the feature extraction process through an improved structure, effectively capturing sparse features and detailed information in the image.

By integrating dilated convolution, windowing operations, and residual connections, the DWRB network significantly enhances computational efficiency and model training effectiveness while maintaining high-resolution and multi-scale feature extraction. Dilated convolution expands the receptive field of the convolutional kernel, enabling the capture of broader context information without increasing computational complexity. The windowing operation divides the input feature map into multiple smaller windows, effectively processing high-resolution images while preserving fine-grained feature information. By introducing skip connections, residual connections facilitate smoother gradient backpropagation, thus alleviating the problem of vanishing gradients and enhancing deep network training. This design improves the network’s training stability, enabling it to better learn complex features.

The DWRB network effectively extracts and fuses multi-scale contextual information through a two-step process to generate a comprehensive feature map. The first step involves creating residual features from the input feature map, referred to as regional residuals. During this phase, regional features of different scales are extracted, and a series of concise feature maps are generated by combining the batch normalization (BN) layer and the 3×3 convolutional layer with the ReLU activation function.

In the second step, the network structure incorporates three branches that downsample the feature maps extracted in the first step to capture multi-scale contextual information. The output feature maps from each branch are then aggregated and concatenated, followed by batch normalization (BN) processing [32]. Next, pointwise convolution is employed to combine the feature maps, forming the final residual feature. Finally, these residual features are added back to the input feature map to construct a more powerful and comprehensive feature representation.

In the three branches of the DWRB network, we have specifically designed the dilated convolution and DRB modules. The dilated convolution is utilized to expand the receptive field, enabling the model to better comprehend both global and local features in the image and effectively capture multi-scale contextual information. In the DRB module, we introduce parallel dilated convolution in addition to using large-kernel convolution. Based on the theory of structural reparameterization, the entire module can be equivalently transformed into a large kernel convolution. This is because the combination of small-kernel convolution and dilated convolution is equivalent to the combination of large-kernel convolution and non-expansive convolution. This design enhances the model’s efficiency in capturing sparsely distributed features in space, thus improving its perception of complex patterns. We integrated the DWRB network module with the C2f module of YOLOv8 to design the C2f_DWRB network structure, as illustrated in Figure 3.

### 3.4. SADown Structure

When dealing with blurred images, the ordinary convolution in the backbone network extracts features using convolution steps or pooling layers, which can lead to the loss of fine-grained information and inefficiencies in feature representation learning. This occurs because traditional convolution operations and pooling processes may not effectively capture key details in blurred images, thus affecting the final feature representation capabilities. To address this issue, we designed the SADown module, as depicted in Figure 4. The SADown module comprises three independent branches: average pooling [33], dilated convolution [34], and SPD-Conv [35]. Each branch utilizes different operations during feature extraction to preserve fine-grained information and enhance the effectiveness of feature representation learning.

To smooth the feature map, reduce its size, and retain as much information as possible, we employ average pooling with a kernel size of 2, a stride of 1, and zero padding. This pooling configuration helps mitigate the impact of noise in practical applications, retain important details, and achieve effective downsampling without significantly reducing the feature map size. By applying local mean calculations, this method smooths the feature map while preserving key feature information, thereby enhancing the quality of feature representation.

The traditional convolution operations typically reduce the resolution of the feature map through downsampling, which can lead to information loss. To address this, we use dilated convolution, which inserts spaces between the elements of the convolution kernel, allowing the receptive field to expand without increasing the number of parameters. This approach enables the model to acquire a larger receptive field, thereby more effectively capturing both global and local features in the image. By enhancing the ability to gather a wide range of contextual information, dilated convolution mitigates the information loss associated with traditional convolution and improves the comprehensiveness and accuracy of feature extraction.

The SPD-Conv (Spatial-to-Depth Convolution) consists of two components: the Space-to-Depth (SPD) layer and the Stepless Convolution (Conv) layer. The SPD layer converts spatial information into the depth (channel) dimension by rearranging the elements of the feature map, thereby achieving downsampling without information loss. This method preserves all information within the channel dimension, avoiding the information loss typically associated with traditional downsampling techniques.

After the SPD layer, the SPD-Conv convolution employs a convolutional layer with a stride of 1 to further process the feature map. This design allows the network to refine feature representations using learnable parameters while preserving spatial information. As a result, it enhances the model’s ability to handle small objects and low-resolution images more effectively.

In general, for a given feature map *X*, the sub-feature map $f_{x,y}$ is derived from all regions in the feature map X(i,j), where *i* and *j* are determined by scaling factors after increasing *x* and *y*, respectively. Consequently, each sub-feature map downsamples the feature map *X* by a scaling factor. For instance, with a scaling factor of 2, if the input image size is (S,S,C), it can be divided into four sub-feature maps, each of size (S/2,S/2,C), using the slicing method described in Formula (1). These four sub-feature maps are then concatenated into a new feature map of size (S/2,S/2,4C) along the channel dimension. Finally, a convolutional layer is applied to generate an output feature map of size (S/2,S/2,C).
(1)$$\begin{array}{cc} f_{0,0} & =X[0:S:2,0:S:2],\\ f_{1,0} & =X[1:S:2,0:S:2],\\ f_{0,1} & =X[0:S:2,1:S:2],\\ f_{1,1} & =X[1:S:2,1:S:2]. \end{array}$$

Finally, we perform a concatenation operation on the feature maps extracted from different branches. By incorporating multiple branches, the model extracts features from various perspectives and methods. Each branch employs different convolution kernels, structures, and parameter settings to capture a diverse range of data. By concatenating these feature maps, the model can comprehensively leverage more feature information, thereby enhancing feature diversity and improving recognition capabilities.

### 3.5. LASPPF Structure

In the Spatial Pyramid Pooling-Fast (SPPF) structure [36], features at different spatial scales are captured through pooling operations at various scales. Typically, different sizes of convolution kernels are employed for these pooling operations to generate feature maps at multiple scales. These feature maps are then concatenated to create a comprehensive feature map containing multi-scale information. However, because SPPF relies solely on maximum pooling, it can lose some global feature information and fail to capture crucial details. To address this issue, we integrate Large Separable Kernel Attention (LSKA) with SPPF, resulting in the design of the LASPPF structure, as illustrated in Figure 5.

LSKA enhances the feature extraction process by decomposing traditional two-dimensional convolution kernels into two one-dimensional convolution kernels [37]. Specifically, LSKA breaks down a large two-dimensional convolution kernel into horizontal and vertical one-dimensional convolution kernels. This decomposition generates an initial attention map, enabling the model to focus more on critical areas in the image. Additionally, using decomposed 1D convolution kernels significantly reduces the number of parameters, thereby lowering computational complexity. This approach is particularly effective for large-scale convolution kernels, as it minimizes memory usage and computational costs. By employing this decomposition strategy, LSKA can greatly reduce computational and memory burdens while maintaining high performance, thus improving the model’s efficiency and scalability.

When decomposing a 2D convolution kernel of size k×k into two 1D kernels of size k×1 and 1×k, the computational complexity is reduced from
$$O(k^{2}\times C_{in}\times C_{out}\times H\times W)$$
to
$$O(k\times C_{in}\times C_{out}\times H\times W),$$
where $C_{in}$ and $C_{out}$ are the input and output channels, and H×W is the spatial resolution of the feature map. This decomposition reduces the computational cost by approximately *k* times. For instance, with a typical kernel size of k=7, the computational savings can be nearly 7-fold. Moreover, this reduction is crucial when working with large-scale convolution kernels, as it significantly lowers the resource demands while preserving the effectiveness of the attention mechanism.

When performing convolution operations, LSKA first applies a one-dimensional convolution kernel for horizontal convolution on the input feature map and then uses another one-dimensional convolution kernel for vertical convolution. After generating a preliminary attention map, LSKA further refines the features using spatial dilated convolutions with various dilation rates. These dilated convolutions expand the receptive field without increasing computational cost, allowing for the capture of a broader range of contextual information. By separately performing horizontal and vertical convolutions, LSKA processes image features in more detail, enhancing the model’s understanding of spatial relationships within images. Following these convolution operations, LSKA fuses the extracted features through a final convolution layer (1×1 conv) to produce the final attention map. This attention map is then element-wise multiplied with the original input feature map, weighting each element according to the attention map values. This process highlights important features and suppresses less relevant ones.

By integrating feature pyramid networks, kernel decomposition, and concatenated convolution strategies, LASPPF utilizes large-scale separable convolution kernels and spatial dilated convolution to capture extensive contextual information from images. The structure generates an attention map that weights the original feature map, enhancing the network’s focus on crucial features and improving model performance. Specifically, LASPPF employs a feature pyramid network (FPN) to combine information across different scales. It uses kernel decomposition to decrease computational complexity while preserving a broad receptive field. Spatial dilated convolution further expands the receptive field, enhancing the model’s ability to capture both global and local features. The final attention map, through element-wise multiplication with the original feature map, highlights significant features and suppresses less important ones. This approach reduces computational and memory costs while maintaining effective image processing capabilities, particularly with large-scale convolution kernels and complex image data.

## 4. Experiment and Results

In this section, we detail the dataset, experimental environment, and parameter configuration utilized in our study, along with the evaluation metrics employed. To validate the effectiveness and feasibility of our proposed model, we conducted both ablation and comparative experiments on two datasets. Ablation Experiments: These experiments are designed to assess the contribution of individual model components to overall performance. By systematically analyzing the impact of different components, we aim to elucidate their specific effects on model improvement. Comparative Experiments: These experiments are conducted to benchmark our model against existing methods. We compare the performance of various models on the same task to demonstrate the advantages and superiority of our approach.

### 4.1. Dataset Introduction

This paper uses two public datasets, GC10-DET dataset [38] and NEU-DET dataset.

GC10-DET is a real industrial steel surface defect detection dataset, which contains 10 types of defects, as shown in Figure 6, namely, punching, weld, crescent gap, water spot, oil spot, silk spot, inclusion, rolling pit, crease, and waist crease. Because some samples in the dataset are unlabeled and mislabeled, after manual screening and preprocessing, 2294 defect images with a size of 2024×1000 were obtained. Then, according to the ratio of 8:1:1, 1836 images in the training set, 229 images in the validation set and 229 images in the test set were randomly divided.

NEU-DET is a dataset of metal surface defects published by Northeastern University. There are 1800 images of 200 × 200, as shown in Figure 7. The defect types include rolled oxide skin (RS), plaque (Pa), cracking (Cr), pitting surface (PS), inclusions (In) and scratches (Sc). According to the ratio of 8:1:1, 1440 images of the training set, 180 images of the verification set and 180 images of the test set were randomly divided.

### 4.2. Experimental Environment and Parameter Setting

In this paper, the DSL-YOLO model is trained on the GPU for 200 rounds, the batch size is 8, and the initial learning rate is 0.01. The performance of the model is evaluated by mean Average Precision (mAP), model parameters, and calculation amount. The basic experimental environment is shown in Table 1.

### 4.3. Evaluation Metrics

This paper analyzes the experimental results accuracy indicators using Precision, Recall, and mean Average Precision (mAP), Parameters, and floating-point operation times GFLOPs. The relevant formulas are as follows:(2)$$P=\frac{\mathrm{TP}}{\mathrm{TP}+\mathrm{FP}}$$
(3)$$R=\frac{\mathrm{TP}}{\mathrm{TP}+\mathrm{FN}}$$
(4)$$\mathrm{AP}=\int_{0}^{1}P\left(R\right)\;dR$$
(5)$$\mathrm{mAP}=\frac{1}{n}\sum_{i=1}^{n}\mathrm{AP}\left(i\right)$$

Among them, TP (True Positive) represents the number of positive classes predicted as positive classes, that is, the number of positive samples that are correctly predicted. FP (False Positive) represents the number of negative classes predicted as positive classes, that is, the number of negative samples mispredicted. FN (False Negative) represents the number of positive classes predicted as negative classes, that is, the number of positive samples mispredicted. Precision refers to the proportion of the actual positive cases in all the samples predicted as positive by the model [39]. It measures the accuracy of the model in positive prediction. Recall refers to the proportion of samples in which the model is successfully predicted to be positive in all samples that are actually positive [40]. It measures the model ’s ability to identify positive examples. The mean Average Precision (mAP) averages the AP values of all categories [41]. AP can reflect the accuracy of each category prediction, and mAP is the average of APs of all classes to reflect the accuracy of the whole model.

### 4.4. Ablation Experiments and Analysis

To assess the effectiveness of the proposed improvements, YOLOv8 is utilized as the baseline model. Subsequently, the C2f_DWRB structure, SADown structure, and LASPPF structure are sequentially incorporated. To evaluate the model’s robustness and generalization capabilities, ablation experiments were conducted on the GC10-DET and NEU-DET datasets. The results for the GC10-DET dataset are presented in Table 2, while the results for the NEU-DET dataset are shown in Table 3.

From Table 2, we can see that when using the baseline model alone, the mAP50 is 68.30%, the Parameters are 3.01 M, and the FLOPs are 8.1 G. When each module is applied to the baseline model, each improvement improves the detection performance to varying degrees. The C2f_DWRB structure can improve the model ’s ability to extract small targets and occluded targets and capture detailed information, and can effectively extract sparse features distributed in space. By introducing the C2f_DWRB structure, mAP50 is increased to 69.20%, the number of parameters is reduced to 2.71 M, and the FLOPs are reduced to 7.70 G. The performance of the model is improved and the accuracy of the detection is improved. The SADown structure is introduced into the model, which can better learn the defect characteristics of small targets and improve the feature representation of the model, so as to improve the performance of the model. The mAP50 is increased to 70.40%, the number of parameters is increased to 3.94 M, and the amount of calculation is increased to 10.00 G. In order to further improve the multi-scale feature extraction ability of YOLOv8 and capture the key information (such as edge, texture and shape) features of the image, we introduce the LASPPF structure, the mAP50 is increased to 72.50%, and the number of parameters is increased to 4.21 M. The calculation amount is increased to 10.30 G.

In industrial applications, real-time requirements necessitate that systems complete tasks within stringent time constraints to ensure the continuity and safety of production processes. Despite the increased computational burden associated with the integration of the SADown and LASPPF modules, the enhanced models remain capable of processing data within the specified time limits, with relatively modest computational resources, thus satisfying the real-time demands of industrial settings.

According to the data in Table 3, the performance index of the baseline model is as follows: mAP50 is 76.40%, the parameter amount is 3.01 M, and FLOPs are 8.1 G. After introducing the C2f_DWRB structure, the mAP50 of the model is increased to 77.70%, the parameter amount is reduced to 2.71 M, and the FLOPs are reduced to 7.70 G. This shows that the C2f_DWRB structure improves the detection accuracy while reducing the computational complexity and memory requirements of the model. After further introducing the SADown structure, mAP50 is further increased to 78.40%. This improvement shows that the SADown structure is superior in extracting fine-grained features and dealing with image blur problems, which further improves the detection performance of the model. After introducing the LASPPF structure, the mAP50 of the model was increased to 79.00%. The LASPPF structure effectively captures multi-scale context information and enhances the model ’s attention to important features by combining large kernel separable convolution and spatial dilated convolution. This improvement shows that our module can significantly improve performance on both datasets, verifying the robustness and generalization ability of the model.

To further validate the effectiveness of our proposed modules, we conducted heatmap visualizations on two datasets, as shown in Figure 8. The heatmaps provide a visual representation of the model’s focus on different regions and highlight variations in feature extraction. The results indicate that the optimized model exhibits increased attention to detail and key defect areas while effectively suppressing background noise. These findings confirm the advantages of our modules in enhancing the model’s detection performance and improving its focus on target regions, especially in the localization and identification of blurred images and minor defects.

### 4.5. Comparative Experiments and Analysis

To further validate the effectiveness and feasibility of our proposed model, we conducted a comparative analysis using two datasets. We compared our model against several widely used object detection algorithms, including YOLOv3, YOLOv5 [42], YOLOv7 [43], and YOLOv8 [44]. The experimental results are presented in Table 4 and Table 5. These comparisons not only demonstrate our model’s performance across different datasets but also underscore its advantages in detection accuracy, speed, and model complexity.

The experimental results demonstrate that the proposed DSL-YOLO model exhibits exceptional performance on both datasets. Compared to traditional detection algorithms, DSL-YOLO outperforms in both accuracy and parameter efficiency. As shown in Table 4 and Table 5, the average mAP of our DSL-YOLO model is 72.5% on the GC10-DET dataset and 79% on the NEU-DET dataset, which represents an improvement of 4.2% and 2.6% over the baseline model, respectively. These results indicate that our model excels in capturing detailed information and handling complex backgrounds, effectively addressing the issues of inadequate feature extraction and context information loss present in traditional algorithms.

In summary, the comparative experiments thoroughly demonstrate the effectiveness and superiority of our proposed DSL-YOLO model for metal surface defect detection. The model exhibits outstanding performance and adaptability on public datasets, highlighting its broad application potential.

To provide a more intuitive reflection of our model’s detection performance, we qualitatively analyzed both the DSL-YOLO model and the baseline YOLOv8n model on the test sets from the GC10-DET and NEU-DET datasets. The results are illustrated in Figure 9 and Figure 10. In these figures, the first column displays the test results of the baseline YOLOv8n model, while the second column shows the detection results of the DSL-YOLO model.

The visual results in Figure 9 and Figure 10 clearly demonstrate the superior performance of our model in handling missed detections and false detections. The DSL-YOLO model effectively identifies and locates metal surface defects with high precision, significantly reducing the rate of errors compared to the baseline. These visualizations highlight the model’s ability to accurately detect defects even in complex scenarios, reinforcing its robustness and reliability in practical applications.

## 5. Conclusions

In this paper, we propose a new DSL-YOLO detection algorithm for metal surface defect detection. This algorithm successfully achieves an optimal balance between detection speed and accuracy, enabling rapid and precise identification of metal surface defects.

To address the limitations of the C2f module in YOLOv8 for detecting small and occluded targets, we propose the C2f_DWRB structure. This modification significantly enhances the model’s ability to capture detailed information and improves the extraction of sparse features in space. Recognizing that standard convolution in the backbone network can lead to the loss of fine-grained information and insufficient feature learning, particularly in blurred images or when processing small objects, we have designed the SADown structure to improve the model’s feature representation capabilities. Additionally, to further enhance multi-scale feature extraction and capture critical image information (such as edges, textures, and shapes) while keeping computational and memory requirements within a reasonable range, we introduce the LASPPF structure.

These improvements significantly enhance the model’s accuracy and efficiency, demonstrating the effectiveness of our approach in metal surface defect detection. Through ablation experiments and comparative analysis on two public datasets, GC10-DET and NEU-DET, we showcase the model’s excellent generalization ability and robustness. The model shows substantial improvements in reducing missed and false detections, with its defect detection and localization capabilities further validated through visual analysis. These results indicate that the DSL-YOLO algorithm proposed in this paper not only improves detection accuracy but also exhibits strong adaptability and stability in practical applications.

In summary, the enhanced DSL-YOLO model demonstrates significant performance improvements in metal surface defect detection tasks. The model excels not only in accuracy and recall but also in model complexity, making it well-suited for various resource-constrained application scenarios. Future work will focus on further enhancing detection accuracy and exploring the model’s deployment and application on edge and embedded devices, aiming to facilitate its widespread use in industrial settings.

## Figures and Tables

**Figure 1 sensors-24-06268-f001:**
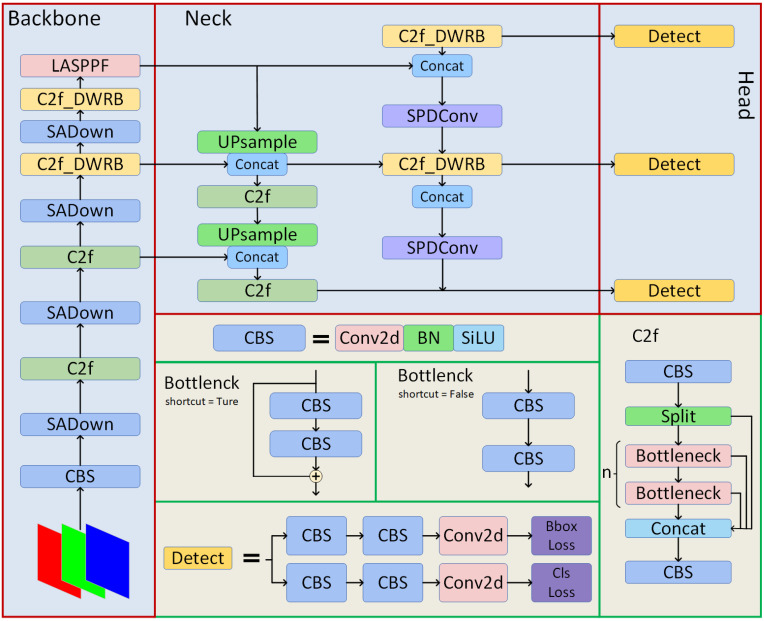
The structure of the DSL-YOLO network.

**Figure 2 sensors-24-06268-f002:**
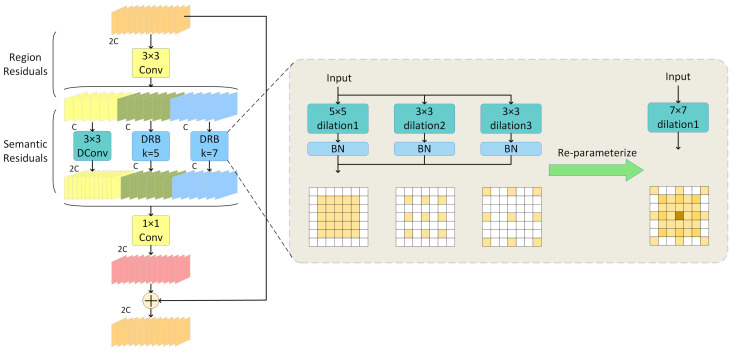
The structure of the DWRB network.

**Figure 3 sensors-24-06268-f003:**
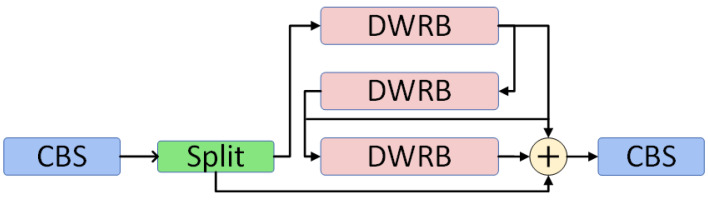
The structure of the C2f_DWRB network.

**Figure 4 sensors-24-06268-f004:**
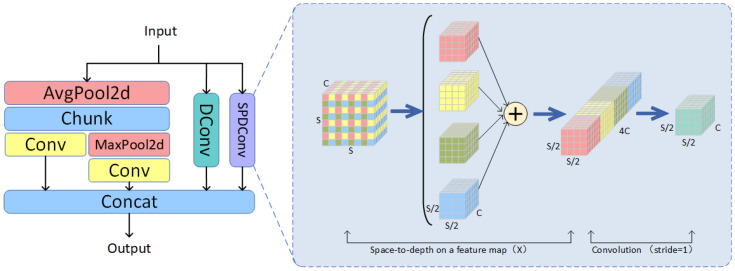
The structure of the SADown network.

**Figure 5 sensors-24-06268-f005:**
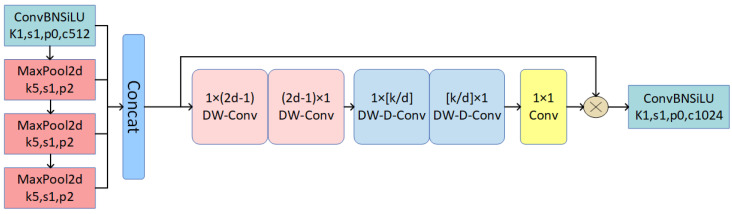
The structure of the LASPPF network.

**Figure 6 sensors-24-06268-f006:**
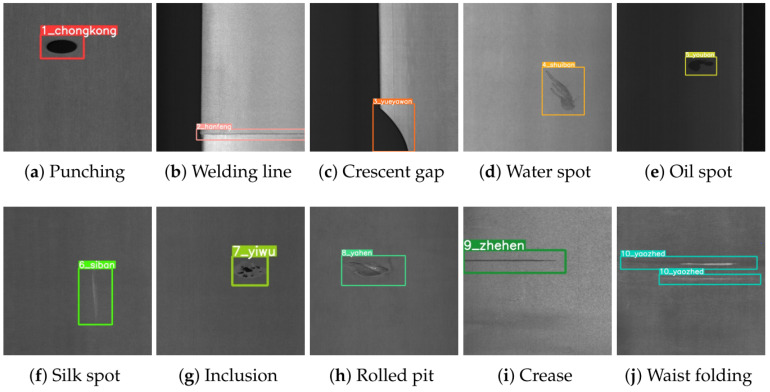
An example of the GC10-DET steel strip surface defect dataset.

**Figure 7 sensors-24-06268-f007:**
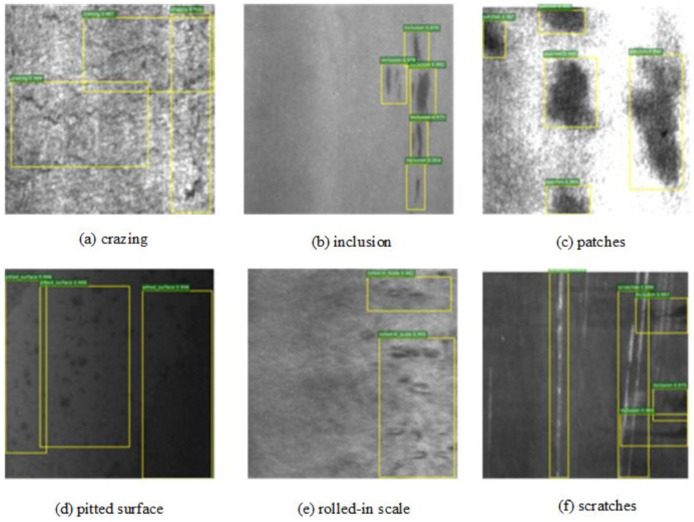
An example of the NEU-DET steel strip surface defect dataset.

**Figure 8 sensors-24-06268-f008:**
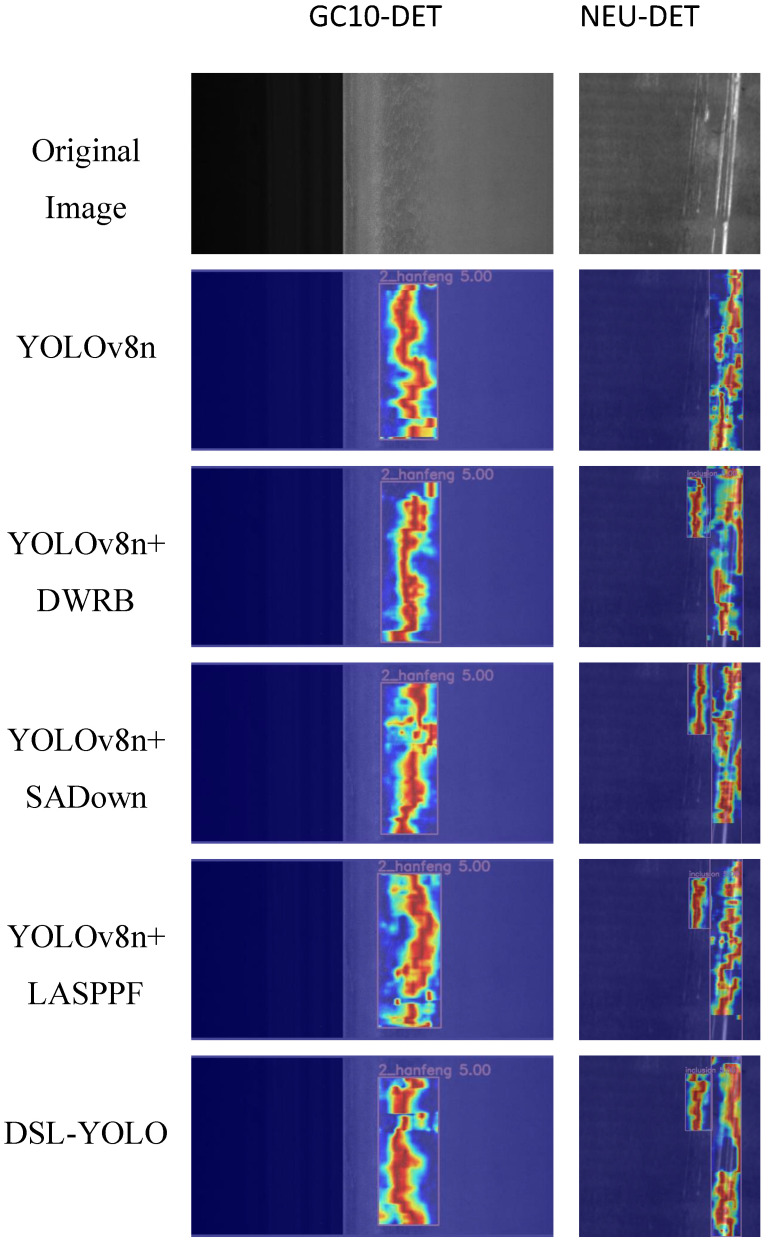
Comparison of heatmap visualization results between the GC10-DET and NEU-DET datasets.

**Figure 9 sensors-24-06268-f009:**
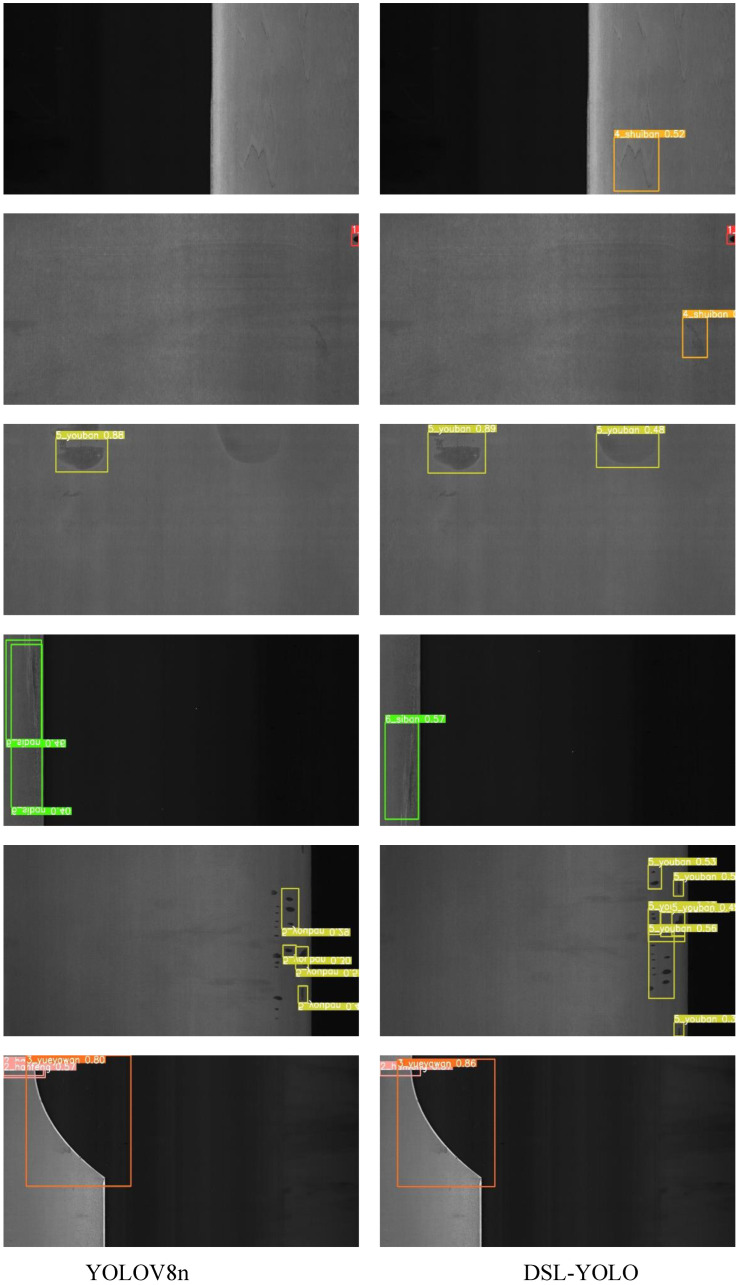
Comparison of visualization results on the GC10-DET dataset.

**Figure 10 sensors-24-06268-f010:**
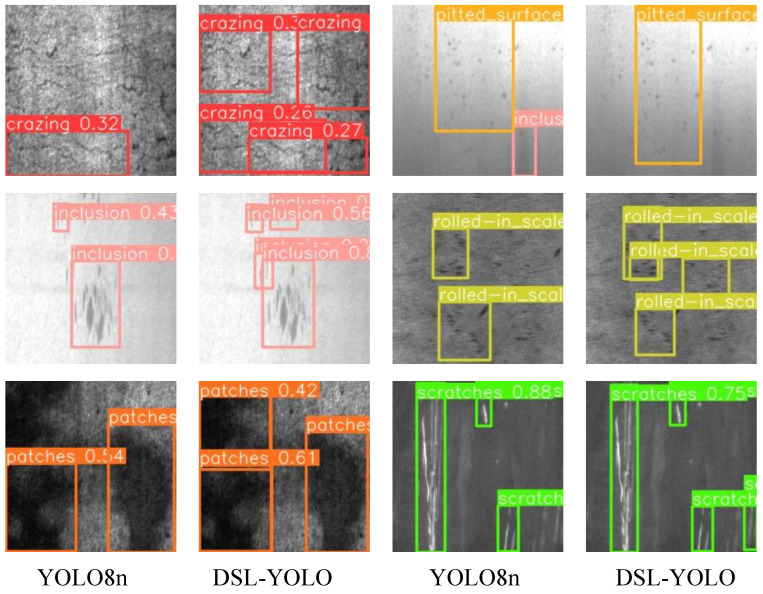
Comparison of visualization results on the NEU-DET dataset.

**Table 1 sensors-24-06268-t001:** Experiment basic environment configuration.

Parameters	Experimental Configuration
Operating system	Ubuntu 20.04.1 LTS
CPU	Intel(R) Xeon(R) CPU E7-4809 v3 @ 2.00 GHz
GPU	NVIDIA GeForce RTX 2080 Ti
Display memory	11 g
Random access memory	32 g
Cuda environment	12.2
Deep learning framework	Pytorch2.2
Programming software	Python3.10

**Table 2 sensors-24-06268-t002:** Results of ablation experiments on dataset GC10-DET.

Model	YOLOv8n	C2f_DWRB	SADown	LASPPF	mAP50/%	Parameters/M	GFLOPs/G
Model_1	*√*				68.3	3.01	8.1
Model_2	*√*	*√*			69.2	2.71	7.7
Model_3	*√*		*√*		70.0	4.23	10.5
Model_4	*√*			*√*	68.9	3.28	8.3
Model_5	*√*	*√*	*√*		70.4	3.94	10.0
Model_6	*√*	*√*		*√*	71.7	2.94	7.9
Model_7	*√*		*√*	*√*	70.5	4.51	10.7
Ours	*√*	*√*	*√*	*√*	72.5	4.21	10.3

**Table 3 sensors-24-06268-t003:** Results of ablation experiments on dataset NEU-DET.

Model	YOLOv8n	C2f_DWRB	SADown	LASPPF	mAP50/%	Parameters/M	GFLOPs/G
Model_1	*√*				76.4	3.01	8.1
Model_2	*√*	*√*			77.7	2.71	7.7
Model_3	*√*		*√*		77.8	4.23	10.5
Model_4	*√*			*√*	76.8	3.28	8.3
Model_5	*√*	*√*	*√*		78.4	3.94	10.0
Model_6	*√*	*√*		*√*	76.1	2.98	7.9
Model_7	*√*		*√*	*√*	78.7	4.51	10.7
Ours	*√*	*√*	*√*	*√*	79.0	4.21	10.3

**Table 4 sensors-24-06268-t004:** Results of comparison experiments on dataset GC10-DET.

Model	Image Size	Precision%	Recall%	mAP50/%	Parameters/M	GFLOPs/G
YOLOv3	640 × 640	68.9	52.2	61.8	61.55	155.3
YOLOv5n	640 × 640	64.9	68.1	68.3	1.77	4.2
YOLOv5s	640 × 640	76.4	65.5	69.9	7.02	15.8
YOLOv7	640 × 640	65.3	62.1	69.4	37.22	105.2
YOLOv7_tiny	640 × 640	59.0	63.8	62.6	6.03	13.2
YOLOV8n	640 × 640	61.9	66.5	68.3	3.01	8.1
Ours	640 × 640	70.5	67.8	72.5	4.21	10.3

**Table 5 sensors-24-06268-t005:** Results of comparison experiments on dataset NEU-DET.

Model	Image Size	Precision%	Recall%	mAP50/%	Parameters/M	GFLOPs/G
YOLOv3	640 × 640	72.6	68.5	73.3	61.55	155.3
YOLOv5n	640 × 640	66.8	70.8	72.3	1.77	4.2
YOLOv5s	640 × 640	72.8	71.3	75.8	7.02	15.8
YOLOv7	640 × 640	64.8	74.6	75.1	37.22	105.2
YOLOv7_tiny	640 × 640	80.8	65.1	72.9	6.03	13.2
YOLOV8n	640 × 640	72.0	71.7	76.4	3.01	8.1
Ours	640 × 640	73.4	75.2	79.0	4.21	10.3

## Data Availability

Publicly available datasets were analyzed in this study. These data can be found here: https://github.com/lvxiaoming2019/GC10-DET-Metallic-Surface-Defect-Datasets (accessed on 26 April 2023). http://faculty.neu.edu.cn/songkechen/zh_CN/zdylm/263270/list/index.htm (accessed on 26 April 2023).

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
