# Peer review of "Research on a Metal Surface Defect Detection Algorithm Based on DSL-YOLO"

_sensors, 2024, doi:10.3390/s24196268_

Round 1

Reviewer 1 Report

Comments and Suggestions for Authors

1. The introduction section should be strengthened. The necessary background information and necessity of the research are needed.

2. The conclusion in the related works section cannot be supported by Ref. 17-21. 

3. In Figure 2., the + sign means add or concat? This figure lacks some signs to mark each step and corresponding module names.

4. In Figure 3., the + sign means add or concat?

5. In section 3.5, how the computational complexity can be reduced through the adaption of LSKA? Could it be quantitatively discussed?

6. Figure 7 is unclear compared with Figure 6.

7. Please explain the 'real-time requirements of the industry' in line 406.

Comments on the Quality of English Language

Moderate editing of English language required. A native speaker may help with this.

Reviewer 2 Report

Comments and Suggestions for Authors

1.      Please give the full name of used abbreviations when first addressing them, such as DSL, DWRB,  LASPPF, etc…

2.      Although some modules are built in module in YOLOv8, it will be more informative to add more details to it including the main functions and more examples to it rather than just one high-level sentence, such as FPN, PAN, etc.

3.      In section 4.4’s result discussion, how to show SADown’s advantage in “improve the feature representation of the mode” and “extracting fine-grained features and dealing with image blur problems” simply by the performance index. It will be beneficial to add more visual evidence to show its advances.

4.      From Table 4 and 5, it can be seen that YOLOV5’s outperforms than others in GC10-DET; while YOLOV5 and YOLOV8 outperforms in NEU-DET, which shows that for different dataset, the model choice might be different. When compare the effectiveness and feasibility of proposed model for different dataset, why choose the same baseline as YOLOv8, rather than the others. 

Reviewer 3 Report

Comments and Suggestions for Authors

This paper introduces DSL-YOLO as a solution to the challenges of low accuracy and slow detection in metal surface defect identification. This work is meaningful. However, several areas require improvement.

1. The introduction and related work lacks some of the defect detection algorithms that have been introduced in the last two years.

2. The research on deep learning-based defect detection encompasses methodologies beyond YOLO, including Transformer models, super-resolution techniques, and others. The manuscript should provide a detailed comparison of the advantages and disadvantages of the current work relative to these alternative approaches.

3. In Method, “YOLOv8n is composed of four main components: input, trunk, neck, and head.” Later on, backbone was used to express it. The article should be expressed uniformly as backbone.

4. The ablation experiments need refinement; the proposed method comprises three modules and should include a series of combined ablation experiments rather than sequentially adding each module for validation.

5. The paper should indicate the resolution of the inputs used for the DSL-YOLO when presenting experiments on the NEU-DET and GC10-DET datasets. The FLOPs measured on different datasets for different resolution inputs are not necessarily consistent.

Round 2

Reviewer 1 Report

Comments and Suggestions for Authors

The authors have considered all my concerns.

Reviewer 3 Report

Comments and Suggestions for Authors

All my concerns are answered and well done.